# What Do We See in Spectra?: Assignment of High-Intensity Peaks of *Cutibacterium* and *Staphylococcus* Spectra of MALDI-TOF Mass Spectrometry by Interspecies Comparative Proteogenomics

**DOI:** 10.3390/microorganisms9061243

**Published:** 2021-06-08

**Authors:** Itaru Dekio, Yuki Sugiura, Susumu Hamada-Tsutsumi, Yoshiyuki Murakami, Hiroto Tamura, Makoto Suematsu

**Affiliations:** 1Department of Biochemistry & Integrative Medical Biology, School of Medicine, Keio University, 35 Shinanomachi, Shinjuku-ku, Tokyo 160-8582, Japan; yuki.sgi@gmail.com (Y.S.); gasbiology@keio.jp (M.S.); 2Department of Environmental Bioscience, Meijo University, 1-501 Shiogamaguchi, Tenpaku-ku, Nagoya, Aichi 468-8502, Japan; tsusumu07@gmail.com (S.H.-T.); hiroto@meijo-u.ac.jp (H.T.); 3Seikakai Mildix Skin Clinic, 3rd Floor, 3-98 Senju, Adachi-ku, Tokyo 120-0034, Japan; bow.t@aol.jp

**Keywords:** *Cutibacterium*, *Staphylococcus*, proteomics, MALDI–TOF mass spectrometry, high-intensity peaks, *S10*-GERMS

## Abstract

Matrix-assisted laser-desorption/ionization time-of-flight (MALDI–TOF) mass spectrometry is a widely used and reliable technology to identify microbial species and subspecies. The current methodology is based on spectral fingerprinting, analyzing protein peaks, most of which are yet to be characterized. In order to deepen the understanding of these peaks and to develop a more reasonable identification workflow, we applied proteogenomic approaches to assign the high-intensity peaks of MALDI–TOF spectra of two bacterial genera. First, the 3–22 kD proteomes of 5 *Cutibacterium* strains were profiled by UPLC–MS/MS, and the amino acid sequences were refined by referring to their genome in the public database. Then, the sequences were converted to *m*/*z* (*x*-axis) values based on their molecular masses. When the interspecies comparison of calculated *m*/*z* values was well-fitted to the observed peaks, the peak assignments for the five *Cutibacterium* species were confirmed. Second, the peak assignments for six *Staphylococcus* species were performed by using the above result for *Cutibacterium* and referring to ribosomal subunit proteins coded on the *S10-spc-alpha* operon (the *S10*-GERMS method), a previous proteomics report by Becher et al., and comprehensive genome analysis. We successfully assigned 13 out of 15 peaks for the *Cutibacterium* species and 11 out of 13 peaks for the *Staphylococcus* species. DNA-binding protein HU, the CsbD-like protein, and 50S ribosomal protein L7/L12 were observed in common. The commonality suggests they consist of high-intensity peaks in the MALDI spectra of other bacterial species. Our workflow may lead to the development of a more accurate species identification database of MALDI–TOF mass spectrometry based on genome data.

## 1. Introduction

Matrix-assisted laser desorption/ionization time-of-flight (MALDI–TOF) mass spectrometry is a widely used technology to identify microbial isolates in clinical settings. The smear of a bacterial isolate on a metal plate is processed with or without an extraction process, followed by crystallization by application of a matrix solution; then, a LASER desorbs the crystallized matrix and sample to form a cloud of positive ions. By the voltage difference of the target plate and the background, these ions are pulled through a vacuum tube. Ions with smaller mass (*m*/*z* value) fly faster to reach the detector at the other end of the tube, finally giving a spectrum of the ionized molecules. The spectral peak pattern of the 2–20 kD range is of low molecular weight proteins, used for species- or subspecies-level identification using a manually created database. In some cases of closely related species like *Streptococcus*, MALDI–TOF can identify bacterial colonies when 16S rRNA gene identification is not successful [1].

Although this fingerprinting method is widely used and considered relatively reliable, misidentification occurs due to inaccuracy of the database derived from the manual collection of spectral data. To improve the identification database, peak calculation should be based on digital analysis based on the genome. However, most of the peaks are unannotated, leaving these spectra as a black box. Theoretically, many of these peaks are believed to be ribosomal subunit proteins [2], and identification protocol using these proteins are being developed (the *S10*-GERMS method) [3]. However, the peaks of the ribosomal subunits are relatively small, and most of the key prominent peaks for species identification remain unidentified [4]. Therefore, an investigation into nonribosomal proteins to identify these peaks is of utmost importance to develop a next-generation database. Teramoto et al. have identified two of a dozen prominent peaks in the spectra of *Cutibacterium acnes* that are used in subspecies discrimination—a CsbD-like protein and a 7 kD antitoxin [4]—and this is the first report of this kind in bacterial MALDI–TOF spectra. The postsource decay (PSD) effect in the commonly used setting for MALDI–TOF species identification, which includes linear mode with α-cyano-4-hydroxycinnamic acid (HCCA/CHCA) as the matrix, is faint [5,6] and can be ignored as an origin of prominent peaks. Therefore, the ions consisting of such peaks should be proton-added protein molecules without a PSD effect. The report of Teramoto et al. proves that at least a part of the proteins, consisting of the prominent peaks, are such crude proton-added nonribosomal proteins.

The reason why most of the annotations are not yet reported may be due to three reasons. First, the proteins encoded in the genome are not always expressed, so a top-down approach from the genome will have difficulty picking up the actual proteins in the specimen. Second, the bottom-up approach is also problematic. Overall, analysis of the protein included in the specimen using MS/MS is troublesome because the sample includes hundreds of proteins, so the peptide production by trypsin digestion results in more than a thousand peptides that cannot be annotated to the correct proteins. Third, the protein database does not include all proteins in a strain; additionally, the amino acid sequence often lacks parts of the proteins.

*Cutibacterium* and *Staphylococcus* are two major bacterial genera residing on human skin. Thus, these are often included in clinical specimens to be sent to hospital laboratories. Genus *Cutibacterium,* with five species including *Cutibacterium acnes* (previously, *Propionibacterium acnes*) [7], was recently coined [8] but is of particular importance as the genus is related to diseases not only of the skin, such as acne vulgaris, but also of internal organs, such as sarcoidosis [9] and prostate cancer [10]. Genus *Staphylococcus* includes 55 species [7], and the genus is also clinically important in various infections. These genera are both Gram-positive and they are included in different major bacterial phyla, namely, *Actinobacteria* and *Firmicutes*. This means the common features of *Cutibacterium* and *Staphylococcus* may be applied to many bacterial species.

In this work, we aim to assign the major peaks of MALDI spectra of these genera by comparing the proteome profile and MALDI spectra using a bottom-up approach with precise techniques, including gel excision and the curation of a database-derived amino acid sequence with the whole genome.

## 2. Materials and Methods

### 2.1. Selection of Bacterial Strains

Eight strains of genus *Cutibacterium* and seven strains of genus *Staphylococcus* were obtained from Japan Collection of Microorganisms (JCM); National Collection of Type Cultures (NCTC), United Kingdom; Culture Collection of University of Goteborg (CCUG), Sweden; and Monash University, Melbourne, Australia (Table 1). The full genome sequences of these strains are available from the DDBJ/EMBL/GenBank public database. For *Cutibacterium* species, strains were grown on trypticase soy agar supplemented with 5% sheep blood (BD, Franklin Lakes, NJ, USA) for 7 days under anaerobic conditions (0% O_2_/10% CO_2_), created in an anaerobic jar with an AnaeroPack sachet (Mitsubishi Gas Chemical, Tokyo, Japan) at 37 °C. Moreover, for *C. acnes* subsp. *acnes* JCM 6425^T^, *C. acnes* subsp. *acnes* JCM 18918, and *C. acnes* subsp. *elongatum* JCM 18919^T^, cultivation was performed under a normal atmosphere (21% O_2_/0.04% CO_2_), and different oxygen pressures for culture were established by the hypoxia workstation Invivo_2_ (Baker Ruskinn, Bridgend, Wales, UK) for 12% O_2_/9% CO_2_, 6% O_2_/15% CO_2_ and 2% O_2_/19% CO_2_. For *Staphylococcus* species, strains were grown on trypticase soy agar supplemented with 5% sheep blood (BD) for 2 days under air (21% O_2_/0.04% CO_2_).

### 2.2. Selection of Genome Data

The whole-genome data for all strains were selected from the database. Some type strains had multiple entries under different strain names, caused by their deposition in multiple culture collections and resource swapping between collections. One of such examples is JCM 6425^T^, the type strain of *C. acnes* and *C. acnes* subsp. *acnes*, listed in fourteen culture collections, according to the JCM online catalog (https://jcm.brc.riken.jp/en/, accessed on 5 June 2021). Its genome data was registered under the strain names of ATCC 6919 (accession numbers CP023676, CP044255, and JNHS01), DSM 1897 (CP025934 and AWZZ01), and NBRC 107605 (AP019723) as of March 2021. In the case of multiple genome submissions under different strain names for one strain, the genome data with the highest status and the latest registration date was selected for analysis (Table 1).

### 2.3. MALDI–TOF Mass Spectrometry

MALDI–TOF mass spectrometry profiles of the strains were obtained using a MALDI Microflex Biotyper (Bruker, Billerica, MA, USA) with parameters as follows: mass range 1960–20,137; ion source 1: 20 kV, 2: 18 kV; lens: 6 kV; detector gain voltage: 2500 V (linear base). For each strain, a cultured colony on the agar plate was taken by a wooden toothpick and smeared onto a target plate (MSP 96 target polished steel BC; Bruker) and fixed with 1 μL HCCA (α-cyano-4-hydroxycinnamic acid) (Bruker). Two hundred laser shots were shot at one point of the target to acquire an accumulated spectrum of 200 spectra, and if the peak height reached a preset threshold, the spectrum was retained. The point in the target was randomly moved. When six accumulated spectra were obtained, all six were added and finalized. The spectra were obtained in triplicate at least, and the representative spectrum with the clearest and rich pattern was selected. The raw spectral data were analyzed using flexAnalysis v3.4 software (Bruker). The spectra data were submitted to jPOST Repository under accession number JPST001137 (PXID: PXD025627) (https://repository.jpostdb.org/entry/JPST001137, accessed on 7 June 2021).

### 2.4. Protein Profiling of 3–22 kD Range to Identify MALDI Peaks

Five *Cutibacterium* strains (*C. acnes* subsp. *acnes* JCM 6425^T^, *C. acnes* subsp. *acnes* JCM 18918, *C. acnes* subsp. *defendens* JCM 6473^T^, *C. acnes* subsp. *elongatum* JCM 18919^T^, and *C. modestum* JCM 33380^T^) were used. A loopful of bacterial cells for each strain was diffused to 2 mL distilled water and lysed by 5 min sonication using an Insonator 201M device (Kubota, Japan). The samples were concentrated under vacuum overnight and then diluted in distilled water to create 40 μL of samples before preservation at −80 °C.

In-gel protein digestion was performed based on the methods of Shevchenko et al. [11] and Misra et al. [12]. First, 10 μg protein samples in 20 μL distilled water were prepared. After being placed in a 70 °C bath for 10 min, the samples and PageRuler Low Range Unstained Protein Ladder (Thermo Fisher Scientific, Waltham, MA, USA) were run on NuPAGE 12% Bis-Tris Gel (1.0 mm, 12 wells) (Thermo Fisher Scientific) with NuPAGE MES SDS Running Buffer (Thermo Fisher Scientific) 200 V for 35 min. Coomassie blue staining was performed using SimplyBlue Safe Stain (Thermo Fisher Scientific), and bands comprising 3–5, 5–12, 12–15, and 15–22 kD were excised by a scalpel (Figure 1). The protein in the gel fragments was destained with 500 μL of 10% acetic acid for 30 min and then washed with distilled water three times. Then, 500 μL of 100% acetonitrile was added and placed at room temperature for 10 min before decantation and 5 min of drying. Reduction: 30–50 μL of 10 mM DTT in 100 mM ammonium bicarbonate was added and incubated at 56 °C for 30 min; 500 μL of 100% acetonitrile was added and placed under room temperature for 10 min before decantation. The DTT amount was determined so that the liquid could cover the gel fragment. Alkylation: 30–50 μL of 55 mM iodoacetamide in 100 mM ammonium bicarbonate was added and placed in the shade for 20 min; 500 μL of 100% acetonitrile was added and placed at room temperature for 10 min before decantation. Digestion: 50–100 μL of 10 mg/mL Trypsin Gold (Promega, Madison, WI, USA) in 100 mM ammonium bicarbonate was added and incubated at 4 °C for 120 min; 10–20 μL of 100 mM ammonium bicarbonate was added so that the gel piece was not exposed to air, and the sample was placed at room temperature overnight. The liquid around the gel was withdrawn and preserved at −80 °C; 100 μL of extraction buffer (1:2 (*vol/vol*) 5% formic acid/acetonitrile) was added to each tube and shaken for 1 hr; 10–20 μL of TFA was added before vortexing and centrifuging at 15,000× *g* for 20 min. The aliquot was withdrawn and preserved at −80 °C.

The samples were then analyzed with the Synapt G2-Si UPLC-MS/MS system (Waters, Milford, MA, USA); 4 uL of the digested materials were analyzed under MS^E^ mode. The raw spectrum data were transferred to ProteinLynx Global Server v3.0.2 software (Waters) for peptide detection and protein identification using the *Cutibacterium* protein database, extracted from the UniProt public database with keywords “*Cutibacterium*”, “sus scrofa trypsin”, “yeast enolase” (yeast enolase was used for the detection check), and “human keratin”.

### 2.5. m/z Value Estimation for Prominent Cutibacterium Peaks

The experimental flow is outlined in Figure 2. The proteins with the masses close to *m*/*z* values of the prominent MALDI peaks of *C. acnes* JCM 6425^T^ (=ATCC 6919^T^) were selected from the protein lists and labeled “candidate proteins”. The amino acid sequences of “candidate proteins” in *C. acnes* JCM 6425^T^ were extracted from the SwissProt database using the keyword search on the UniProt webpage (https://www.uniprot.org, accessed on 5 June 2021). The gene positions and the translated amino acid sequences of these “candidate proteins” in the three *Cutibacterium* type strain genomes (accession numbers CP044255 for *C. acnes* subsp. *acnes* type strain ATCC 6919^T^, CP003084 for *C. acnes* subsp. *defendens* type strain ATCC 11828 ^T^, and LT906441 for *C. granulosum* type strain NCTC 11865^T^) were searched using the NCBI tblastn webtool (https://blast.ncbi.nlm.nih.gov/Blast.cgi, accessed on 5 June 2021). Then, by using the NCBI Sequence Viewer (https://www.ncbi.nlm.nih.gov/projects/sviewer/, accessed on 5 June 2021), the amino acid sequences were checked and curated using the correspondent genome sequences as templates. The masses of these proteins were calculated by using the Protein/Peptide Editor function of MassLynx v4.2 software (Waters) and converted to *m*/*z* values for singly charged ions [M + H]^+^ and doubly charged ions [M + 2H]^+^ using the equations below to check if these values matched the observed MALDI peaks (error range of 500 ppm was allowed). 

For singly charged ions,
*m*/*z*_0_ ([M + H]^+^) = (molecular weight) + 1.01

For doubly charged ions,
*m*/*z*_0_ ([M + 2H]^+^) = { (molecular weight) + 1.01 × 2}/2

Additionally, the *m*/*z* values of the amino acid sequences with N-terminal (post-translational modification) Met or Val removed [3,13] were checked using the equations below based on the difference of the molecular weight without Met or Val, which were −130.49 and −98.12, respectively.

For singly charged ions with N-terminal Met removed,
*m*/*z*-_M_ ([M + H]^+^) = (molecular weight − 130.49) + 1.01

For singly charged ions with N-terminal Val removed,
*m*/*z*-_V_ ([M + H]^+^) = (molecular weight − 98.12) + 1.01

For doubly charged ions with N-terminal Met removed,
*m*/*z*-_M_ ([M + 2H]^+^) = { (molecular weight − 130.49) + 1.01 × 2}/2

For doubly charged ions with N-terminal Val removed,
*m*/*z*-_V_ ([M + 2H]^+^) = { (molecular weight − 98.12) + 1.01 × 2}/2

If either *m*/*z*_0_ or *m*/*z*_-M/-V_ seemed to match a MALDI peak of all the three type strains of *C. acnes* subsp. *acnes*, *C. acnes* subsp. *defendens*, and *C. granulosum,* the same workflow was performed on the other type strain draft genomes (whole-genome shotgun project entries: BAVO for *C. acnes* subsp. *acnes* JCM 18918, BFFM for *C. acnes* subsp. *elongatum* JCM 18919^T^, AGBA for *C. avidum* ATCC 25577^T^, BJEN for *C. modestum* JCM 33380^T^, and LWHO for *C. namnetense* NTS 31307302^T^) to check if the *m*/*z* values also matched MALDI peaks of these five type strains.

### 2.6. m/z Value Estimation for Prominent Staphylococcus Peaks

First, we referred to the assignment result of *Cutibacterium* (7 proteins for 13 peaks) to create a list of ‘candidate proteins’. The UniProt search using keywords as the protein name and ‘*Staphylococcus*’ was performed, and the retrieved amino acid sequence was used as a template for the tblastn search to find a corresponding genome sequence of *S. aureus* DSM 20231^T^ (accession number: CP011526). Then, the DNA sequence was used to curate the template amino acid sequence. The *m*/*z* calculation was done, as in the above *Cutibacterium* section, to see if the *m*/*z*_0_ or *m*/*z*_-M/-V_ values of each ‘candidate protein’ matched the observed peaks in the *Staphylococcus* MALDI spectrum. However, this workflow could assign only four peaks derived from two proteins and left the majority of the peaks unassigned. Secondly, we used the 2D gel proteomics results by Becher et al. [14] (Appendix A of this reference, 40 proteins in the <20 kD area) to create another list of ‘candidate proteins’, and the same workflow was applied. This workflow could assign only one peak derived from one protein. Thirdly, we created a list of ribosomal subunit proteins coded on the *S10-spc-alpha* operon (24 proteins) [3] to serve as another list of ‘candidate proteins’ and performed the same analysis. Again, this workflow could assign only one peak derived from one protein.

Finally, we searched the *S. aureus* DSM 20231^T^ genome to find possible proteins to form the peaks. The complete genome of the *S. aureus* subsp. *aureus* type strain, DSM 20231^T^ (accession number: CP011526), was used to construct a genome-wide protein molecular weight list. Briefly, the annotated amino acid sequences were extracted, converted to a MultiFasta format, and then subjected to molecular weight calculation using online tools provided by the Belgian Co-ordinated Collections of Micro-organisms (https://www.genecorner.ugent.be/protein_mw.html, accessed on 7 June 2021) and the Swiss Institute of Bioinformatics (https://web.expasy.org/compute_pi/, accessed on 7 June 2021). To visualize the molecular weight distribution, the number of proteins in each kD range was counted. The N-end rule and the addition of a proton were applied to calculate *m*/*z*. Similarly, we also constructed protein molecular weight lists for AYP 1020, DSM 6717^T^, NCTC 12196^T^, ATCC 14990^T^, NCTC 11320^T^, and NCTC 12217^T^ using the complete genomes or sets of whole-genome shotgun scaffolds (accession Nos: CP007601, PPQI01, PPRT01, CP035288, PPQE01, and LS483482, respectively). An alignment of the 5.5 kD hypothetical protein encoded in the genome of DSM 20231^T^, DSM 6717^T^, NCTC 12196^T^, and NCTC 11320^T^ was created using ClustalW 2.0 (GenomeNet, https://www.genome.jp/tools-bin/clustalw, accessed on 7 June 2021). This workflow resulted in five peaks derived from five proteins.

## 3. Results

### 3.1. MALDI–TOF Peak Comparison of Cutibacterium acnes across Different Oxygen Pressures

Three *Cutibacterium acnes* strains were cultured under five different oxygen pressures. The MALDI–TOF peaks of the 7 kD antitoxin and the CsbD-like protein were observed in all spectra, and the 7 kD antitoxin peak height to the CsbD-like protein peak height ratio was lower in lower oxygen pressures (Figure 3A).

### 3.2. MALDI–TOF Mass Spectrometry Profiles of Genus Cutibacterium

The spectra of the eight strains were successfully obtained. The spectra contained 13 prominent peaks, and the *m*/*z* values of the MALDI–TOF peaks were different between strains.

On the other hand, 43 proteins were identified from the excised 1D gels in the 3–22 kD range of *Cutibacterium* extracts. These included 40 nonribosomal proteins and 3 ribosomal subunits. Among the nonribosomal proteins, 25 were with molecular weights less than 16 kD (Appendix A) and, thus, considered a pool of potential “candidate proteins”. By calculating the masses of the protein ions listed in the UniProt database and those without an N-terminus amino acid of methionine or valine (see the example in Figure 3B), the singly charged ions of 6 proteins matched the peaks of multiple strains: Lsr2-like protein -Met [M + H]^+^ at *m*/*z* 11,651.76 for *C. acnes* subsp. *acnes* type strain, 10 kD chaperonin GroS -Val [M + H]^+^ at *m*/*z* 10,497.97, DNA-binding protein HU [M + H]^+^ at *m*/*z* 9589.07, CsbD-like protein -Val/-Met [M + H]^+^ at *m*/*z* 7179.95, 7 kD antitoxin -Met [M + H]^+^ at *m*/*z* 7034.57, and DUF3117 domain-containing protein -Met [M + H]^+^ at *m*/*z* 5712.64. The doubly charged ions of three out of these six proteins were also observed as MALDI peaks of multiple strains, which were Lsr2-like protein -Met [M + 2H]^2+^ at *m*/*z* 5826.39 for *C. acnes* subsp. *acnes* type strain, 10 kD chaperonin GroS -Val [M + 2H]^2+^ at *m*/*z* 5249.49, and DNA-binding protein HU [M + 2H]^2+^ at *m*/*z* 4795.04. Furthermore, using the analysis of genus *Staphylococcus*, as described later, other two peaks were assigned as the singly and doubly charged ions of 50S ribosomal protein L7/L12, -Met [M + H]^+^ at *m*/*z* 13,571.41 and -Met [M + 2H]^2+^ at *m*/*z* 6786.21, respectively (Figure 4, calculation in Appendix A).

These six proteins included two previously reported assignments, which were a CsbD-like protein and a 7 kD antitoxin [3]. Our results were consistent with theirs except for the *m*/*z* value for the 7 kD antitoxin for *C. acnes* subsp. *elongatum* JCM 18919^T^ at 6971.5 (calculated) or 6969.3 (observed) instead of 7004.

On the other hand, we could not characterize two peaks at 15,043 and 7522, even after adding the calculation considering the effects of disulfide binding, methylation, acetylation, and splicing. The *m*/*z* values suggest these peaks are derived from the same protein: 15,043 being a singly charged ion and 7522 a doubly charged ion. These did not appear when we changed the agar plates from blood-containing plates (trypticase soy agar supplemented with 5% sheep blood, BD) to nonblood-containing plates (trypticase soy agar, BD), suggesting the peaks were derived from a blood component. Indeed, our UPLC–MS/MS analysis of *Cutibacterium* species using the whole protein list in the UniProt database resulted in a hemoglobin α chain, whose singly and doubly charged ions have calculated *m*/*z* values close to these observed peaks (Appendix A).

### 3.3. MALDI–TOF Mass Spectrometry Profiles of Genus Staphylococcus

The spectra of the seven strains were successfully obtained. Each spectrum contained 13 high-intensity peaks, and the *m*/*z* values of these peaks were different between strains.

By using the result of genus *Cutibacterium*, four peaks were assigned for two proteins: DNA-binding protein HU ([M + H]^+^ at *m*/*z* 9627.02 and [M + 2H]^2+^ at *m*/*z* 4814.02 for *S. aureus* type strain) and CsbD-like protein ([M + H]^+^ at *m*/*z* 6888.51 and [M + 2H]^2+^ at *m*/*z* 3444.76). By using the protein list of a previous report by Becher et al. [14], one peak was assigned as a singly charged ion derived from a ribosomal subunit protein: 50S ribosomal protein L7/L12 (-Met [M + H]^+^ at *m*/*z* 12,580.36). By using a list of ribosomal subunit proteins coded on the *S10-spc-alpha* operon, one peak was assigned as a singly charged ion derived from a ribosomal subunit protein: 50S ribosomal protein L36 ([M + H]^+^ at *m*/*z* 4306.37). Finally, to identify proteins corresponding to the remaining seven major peaks, we calculated the molecular weight and theoretical *m*/*z* of whole coding sequences annotated in the complete genome of *Staphylococcus aureus* DSM 20231^T^ (see Materials and Methods). The amino acid sequence was available for 2606 coding sequences in the genome. Among these, the molecular weight of 523 proteins was within 3–15 kD, which is the detectable range of MALDI–TOF MS (Figure 5A). As a result of a comprehensive search using the genome-wide protein molecular weight list, we could assign five peaks as singly charged ions of four proteins and one putative protein: 30S ribosomal protein S15 (-Met [M + H]^+^ at *m*/*z* 10,478.07), 30S ribosomal protein S16 (-Met [M + H]^+^ at *m*/*z* 10,104.68), 50S ribosomal protein L32 (-Met [M + H]^+^ at *m*/*z* 6354.36), 5.5 kD hypothetical protein (-Met [M + H]^+^: 5525.00), and glycopeptide resistance-associated protein GraF (-Met [M + H]^+^: 5032.49). These assignments consisted of a total of 11 out of 13 high-intensity peaks (Figure 5B,C; calculation in Appendix A).

On the other hand, two peaks at 15,043 and 7522 were observed in a similar pattern as genus *Cutibacterium*, and we could not identify these peaks. The *m*/*z* values of these peaks seemed conserved across species.

## 4. Discussion

We could successfully assign the majority of the high-intensity peaks of MALDI–TOF spectra for *Cutibacterium* and *Staphylococcus* species by proteogenomic approaches, and this is the first report of such analysis for bacterial MALDI–TOF mass spectrometry. Our methodology includes a bottom-up approach by UPLC–MS/MS analysis, with gel excision and curation of amino acid sequences with genome data, and a top-down approach by protein-coding lesion analysis in a genome. In *Cutibacterium* spectra, only 2 out of 13 peaks were of ribosomal subunit proteins, making a striking contrast with *Staphylococus* spectra, in which 5 out of 11 peaks were of ribosomals. Our findings lead to a potential breakthrough for the development of a more precise and lighter database for MALDI–TOF microbial identification of bacterial species.

### 4.1. Stability of MALDI Major Peaks and Rationale of Methodology

We observed that the height ratio of the two major peaks around 7 kD of MALDI–TOF spectra of *Cutibacterium acnes* showed a linear relationship with different oxygen pressures (Figure 3A). Although the mechanism causing this ratio difference remains unknown, this finding suggests that the peak heights are semiquantitative, and these major peaks should be similarly present in spectra for a strain under different laboratories.

During the assignment process, we noticed several proteins are present at a similar mass range, so that it is often confusing to characterize. The difference of the calculated masses across subspecies/species is helpful in determining the assignment. On the other hand, our genome analysis of the *Staphylococcus aureus* type strain showed that around 50 proteins were coded in a 1 kD window of the 3–15 kD area, where MALDI spectra are detected. As we applied a 500 ppm error range, the difference between calculated and observed *m*/*z* values was within 3.5 for 7 kD and 5 for 10 kD. This means the top-down approach gives several proteins with *m*/*z* values close to an observed peak. Therefore, a multispecies comparison within a genus is crucial for the final verification of the assignment.

### 4.2. Identified Proteins Consisting of Major MALDI–TOF Spectra Peaks for Cutibacterium

We identified six nonribosomal proteins, which were Lsr2-like protein, 10 kD chaperonin GroS, DNA-binding protein HU, 7 kD antitoxin, CsbD-like protein, and DUF3117 domain-containing protein, forming the prominent peaks of *Cutibacterium* spectra. Among these six proteins, two (10 kD chaperonin GroS and DNA-binding protein HU) were related to cell multiplication and one (7 kD antitoxin) to immunity-like protection, while the functions of three (Lsr2-like protein, CsbD-like protein, and DUF3117 domain-containing protein) were unknown. The observation is that at least two of these proteins are related to replication; this may reflect the log phase when the bacterial cells were harvested; 7 kD antitoxin and CsbD-like protein had been identified as consisting of the two useful MALDI-TOF peaks in discriminating *C. acnes* subspecies [15,16], which are at 7033 and 7179 for *C. acnes* subsp. *acnes* type IA_1_ [4].

Additionally, 10 kD chaperonin GroS (GroES) is known to exist in most bacteria and creates a dimer with 60 kD chaperonin GroL (GroEL) to work as a machine to fold other proteins [17]. DNA-binding protein HU is one of the histone-like DNA-binding proteins abundant in all prokaryotes, and it has the ability to stabilize double-stranded DNA against thermal denaturation [18]. The 7 kD antitoxin is speculated as one of several toxin–antitoxin protection system members in *Cutibacterium* species. A UniProt database search with keywords antitoxin and *Cutibacterium* gave at least seven proteins, including HicA-HicB and VapC-VapB toxin–antitoxin pairs, which have been deeply investigated [19]. Unfortunately, the 7 kD antitoxin is yet unnamed and uncoupled. Considering that the masses of these well-known pair proteins were of 7–14 kD in the database but did not form prominent peaks in MALDI spectra, this 7 kD antitoxin is a more dominant member of the toxin–antitoxin system in *Cutibacterium* species that needs further research. Lsr2 is a protein that regulates gene expression in *Mycobacterium tuberculosis* [20]. To understand the role of the Lsr2-like protein we encountered, we checked the amino acid sequence of Lsr2 in *Mycobacterium* and compared this with the Lsr2-like protein of *C. acnes*. The Lsr2-like protein of *C. acnes* type strain DSM 1897^T^ (=JCM 6425^T^) (accession number A0A6B7EXG2) had only 44% similarity to Lsr2 in *Mycobacterium tuberculosis* type strain H37Rv. Due to the relatively low similarity score, we could not judge whether our Lsr2-like protein functions as Lsr2. In a similar search, the CsbD-like protein of *C. acnes* type strain JCM 18918 (accession number W4U2V9) had almost no similarity to CsbD (=ywmG) in *Bacillus subtilis* (accession number P70964), which is a stress response protein related to phosphate starvation [21]. This protein is studied only in *B. subtilis*, and so we have no information on how our CsbD-like protein works. DUF3117 is described in the Pfam protein database (as of Feb 2021) as a protein of unknown function restricted to phylum *Actinobacteria*, and we came to no explanation after an intensive search of the literature as well as of the DUF3117-domain-containing protein we encountered.

### 4.3. Identified Proteins Consisting Major MALDI–TOF Spectra Peaks for Staphylococcus

Many of the MALDI–TOF peaks of genus *Staphylococcus* were derived from ribosomal proteins (5 out of 13 peaks) and proteins described in the *Cutibacterium* section (4 out of 13 peaks; DNA-binding protein and CsbD-like protein). The other two proteins were a 5.5 kD hypothetical protein and glycopeptide resistance-associated protein GraF. The former is a hypothetical small protein consisting of 48 amino acids, and we firstly described its existence from our comprehensive genome analysis. No similar protein has been reported so far, and, thus, no explanation can be made about its function at this stage. The latter is a small protein consisting of 44 amino acids, firstly reported in 2005 as one of the influential genes in decreasing glycopeptide susceptibility and cell wall thickness in *S. aureus* [22]; however, no other literature about this protein follows. Although not characterized further, the peaks corresponding to these two proteins are useful for the identification of bacteria by the MALDI–TOF method.

### 4.4. Commonly Observed Peaks at m/z 15,041/7522 in Both Cutibacterium and Staphylococcus

We observed two peaks at *m*/*z* 15,041 and 7522 in the spectra of the majority of the *Cutibacterium* and *Staphylococcus* strains, and we could not assign proteins to these in our analysis. Considering that the peaks uniformly occur at similar *m*/*z* values across species, we speculated these do not derive from the bacterial strains. We used a trypticase soy agar plate supplemented with 5% sheep blood, and our speculation reached the culture methodology. When we changed the agar plate to a trypticase soy agar plate without blood, these peaks did not appear (Appendix A). Moreover, in the UPLC–MS/MS analysis of *Cutibacterium*, we frequently detected the hemoglobin α chain. Therefore, we speculate that these peaks are derived from an ingredient of the blood carried over from the agar plates, namely, hemoglobin α chain (calculated *m*/*z* for sheep: -Met [M + H]^+^ 15,034.18), with the molecular weight of 15,040 (for singly charged [M + H]^+^, *m*/*z* = 15,041, for doubly charged [M + 2H]^2+^, *m*/*z* = 7522). However, we did not conclude the assignment for two reasons. First, we did not make a comparison of the peak to a standard. Second, *m*/*z* values of the observed peaks were uniformly larger than calculated, about 10 *m*/*z*, and we have no explanation for this discrepancy.

### 4.5. Comparison of MALDI Spectra at Subspecies Level

Several reports have stated that bacterial identification is possible down to subspecies level using MALDI–TOF mass spectrometry. For *Cutibacterium* species, Nagy et al. [15] and ourselves [16,23] have reported that *C. acnes* subspecies can be identified by the *m*/*z* values of two peaks around 7 kD; later, Teramoto et al. showed that these were a CsbD-like protein and a 7 kD antitoxin [4]. Several other species that can be identified at the subspecies level from direct MALDI–TOF MS include *Bifidobacterium animalis* [24], *Mycobacterium abscessus* [25], and *Fusobacterium nucleatum* [26]. However, no such report can be found in *Staphylococcus* species. Our result shows no difference in *m*/*z* values of major peaks is present in two subspecies of *Staphylococcus capitis*, and this indicates that these subspecies cannot be identified by direct MALDI–TOF MS. In this context, *C. acnes* and *S. capitis* subspecies showed a striking contrast, and a precise check of the protein-coding lesions of the genomes may reveal whether subspecies-level identifications can be achieved by MALDI–TOF fingerprinting.

## 5. Conclusions

While using MALDI–TOF MS for bacterial identification, the majority of the high-intensity MS peaks were not identified at all in the fingerprint method; this leads to ambiguity and a large amount of labor for creating a database. The recently developed *S10*-GERMS method [3], which analyses the peaks corresponding to theoretical *m*/*z* values calculated from parts of the genome, could make it possible to identify strains at species, subspecies, or even lower levels. This implies that MALDI–TOF MS bacterial identification, coupled with the bioinformatics approach, will be an important screening tool. Therefore, this study aimed to determine the nature of key high-intensity peaks ubiquitously observed in MALDI–TOF spectra based on assignment and validation by both top-down and bottom-up procedures.

Since the *S10*-GERMS method represents the sequence variations of proteins (not only the house-keeping proteins but also the nonubiquitous proteins), the possibility of the *S10*-GERMS method combined with results from a genomic database will be developed as a highly reliable advanced method for phylogenetic analysis as well as bacterial discrimination at the strain level.

Our result that the high-intensity MS peaks are assignable, and the fact that a part of them are common between genera suggests that a part of high-intensity MS peaks can be calculated from their genomes in any bacterial species/subspecies, paving the way for the development of lighter and easier-to-use identification databases.

## Figures and Tables

**Figure 1 microorganisms-09-01243-f001:**
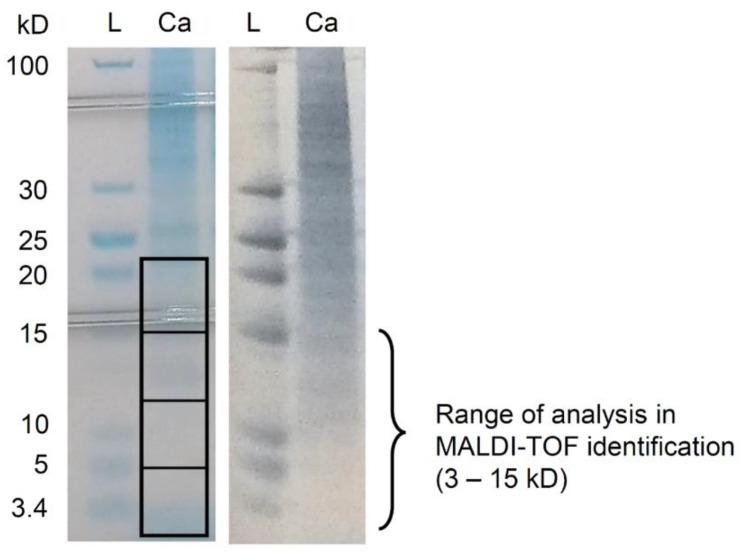
Example of 1D gel excision for proteomics. The material was extracted protein from *Cutibacterium acnes* subsp. *acnes* JCM 6425^T^. Left: Coomasie blue staining with SimplyBlue Safe Stain (Thermo Fisher Scientific). Rectangles indicate excision areas (3–5, 5–12, 12–15, and 15–22 kD). Right: reverse staining with ExStain Reverse (Atto, Tokyo, Japan). L: ladder. Ca: *Cutibacterium acnes*.

**Figure 2 microorganisms-09-01243-f002:**
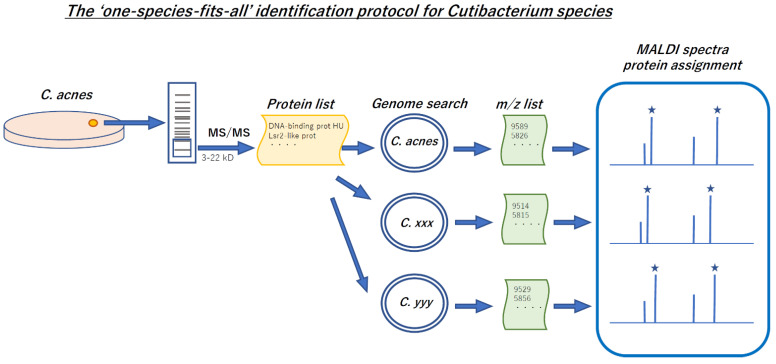
Scheme of the experimental flow for *Cutibacterium* MALDI peaks.

**Figure 3 microorganisms-09-01243-f003:**
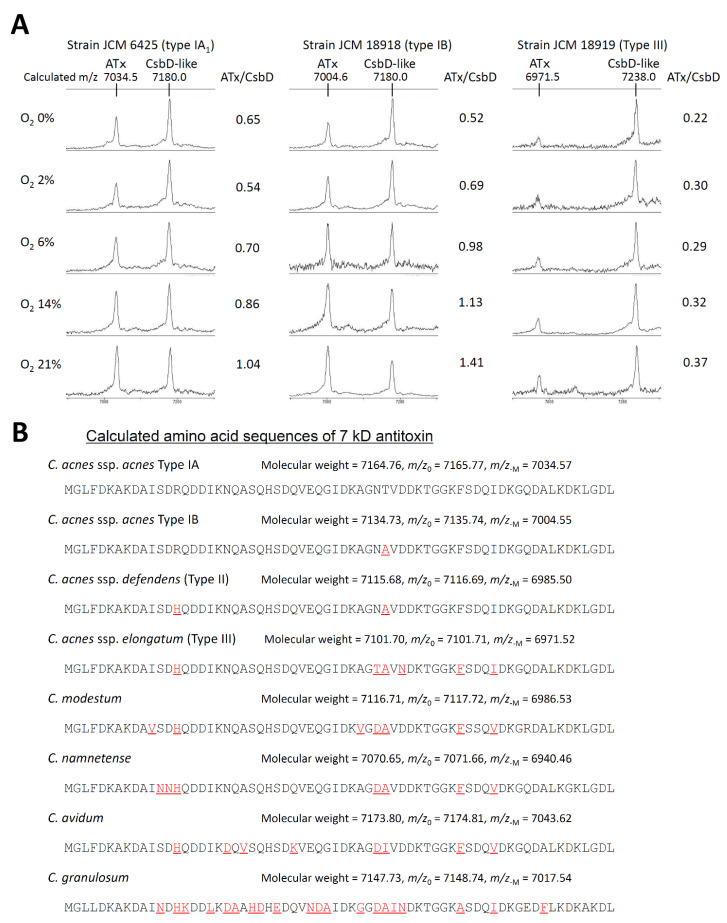
Background data for the MALDI peak assignment. (**A**) 7 kD antitoxin and CsbD-like protein peaks of three *Cutibacterium acnes* strains cultured under different oxygen pressures. ATx: 7 kD antitoxin, CsbD: CsbD-like protein, ATx/CsbD: peak height ratio of 7 kD antitoxin and CsbD-like protein. Note that ATx/CsbD is higher when oxygen pressure is higher, which indicates the positive relationship between peak height and protein amount. (**B**) Comparison of amino acid sequences of 7 kD antitoxin in *Cutibacterium* species and subspecies calculated from their genomes. Highlights with red and underlines indicate sequence differences from *C. acnes* subsp. *acnes* Type IA. The same calculations were also performed on the other ‘candidate proteins’.

**Figure 4 microorganisms-09-01243-f004:**
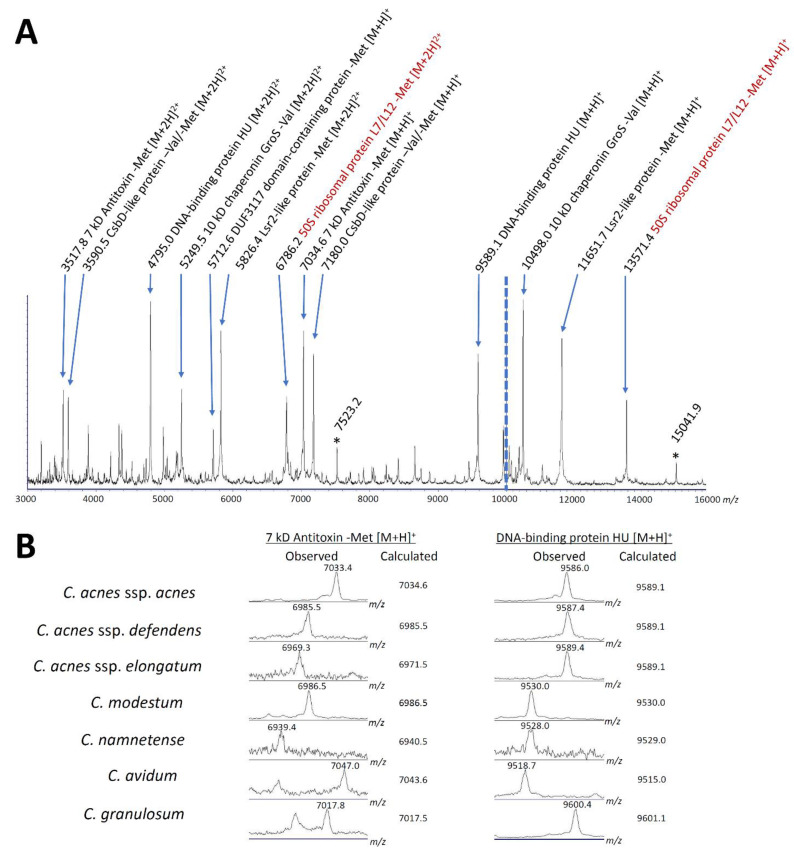
Assignment of prominent MALDI peaks of *Cutibacterium acnes*. (**A**) Peak assignment on the overall spectra of *C. acnes* subsp. *acnes* JCM 6425^T^. Both *x*- and *y*-axes are different between 3000–10,000 *m*/*z* and 10,000–16,000 *m*/*z* ranges. Numbers before protein assignments are calculated *m*/*z* values (observed *m*/*z* values, see Appendix A). Brown letters indicate ribosomal subunit proteins. -Met, -Val: methionine and valine at N-terminus deleted from original translations. * Nutrition-dependent peaks yet to be identified (singly and doubly charged ions of the same protein). (**B**) Examples of assignment by comparing the observed and calculated peaks. The calculation was performed by a bottom-up approach based on UPLC–MS/MS profiling of the proteins.

**Figure 5 microorganisms-09-01243-f005:**
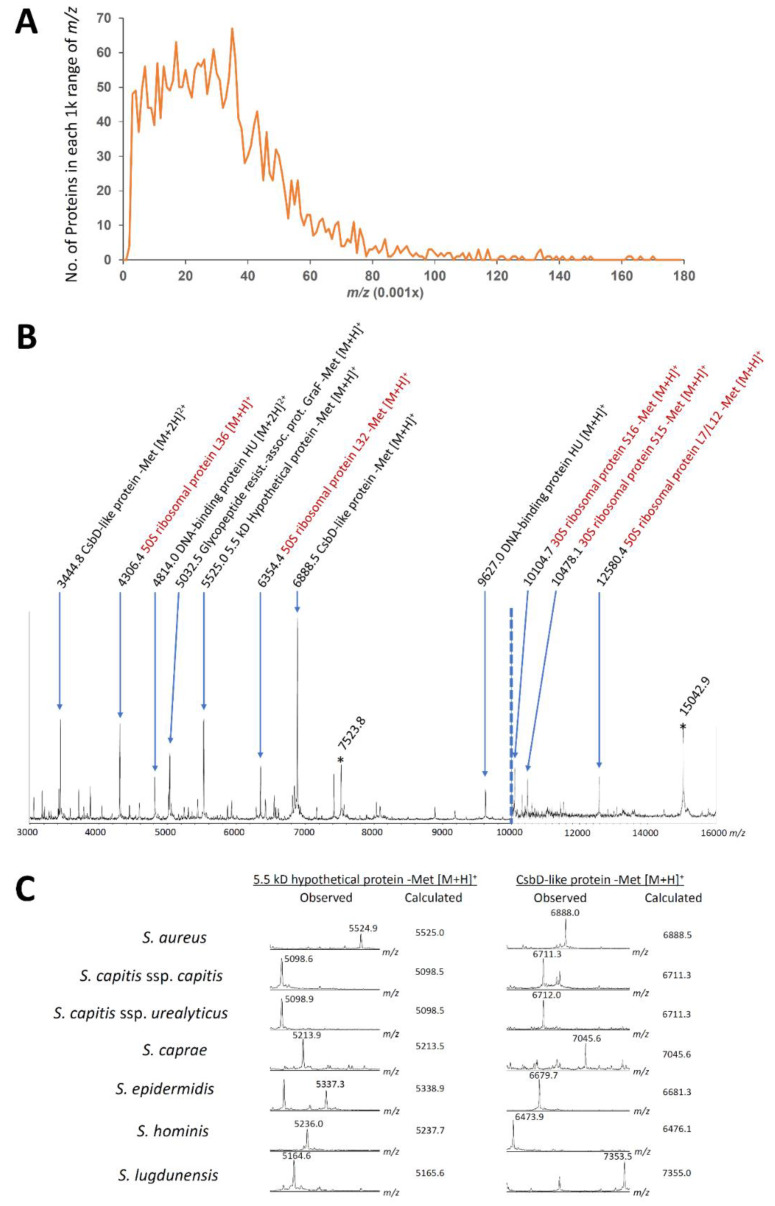
Assignment of prominent MALDI peaks of *Staphylococcus aureus*. (**A**) Number of proteins coded in the *Staphylococcus aureus* DSM 20231^T^ genome. Approximately 40–60 proteins are coded in each 1 kD range. This indicates that the protein assignment can be confirmed when a calculated peak fits an observed peak within the 500 ppm range for multiple species. (**B**) Peak assignment on the overall spectra of *S. aureus* CCUG 1800^T^. Both *x*-axis and *y*-axis are different between 3000–10,000 *m*/*z* and 10,000–16,000 *m*/*z* ranges. Numbers before protein assignments are calculated *m*/*z* values (observed *m*/*z* values, see Appendix A). Brown letters indicate ribosomal subunit proteins. -Met: methionine at N-terminus deleted from original translations. * Nutrition-dependent peaks yet to be identified (singly and doubly charged ions of the same protein). (**C**) Examples of assignment by comparing the observed and calculated peaks. The calculation was performed on *Cutibacterium* results, ribosomal subunit proteins coded on the *S10-spc-alpha* operon, and a 2D analysis report by Becher et al. [11].

**Table 1 microorganisms-09-01243-t001:** Strains used. T after strain number indicates type strains of species/subspecies. * Identical to the used strain (same strain deposited from different culture collections).

Species	Strain Number	Genome Entry
*Cutibacterium acnes* subsp. *acnes*	JCM 6425^T^ (Type IA_1_)	CP044255 (ATCC 6919^T^ *)
	JCM 18918 (Type IB)	BAVO01 (JCM 18918)
*Cutibacterium acnes* subsp. *defendens*	JCM 6473^T^	CP003084 (ATCC 11828^T^ *)
*Cutibacterium acnes* subsp. *elongatum*	JCM 18919^T^	BFFM01 (JCM 18919^T^)
*Cutibacterium avidum*	NCTC 11864^T^	AGBA01 (ATCC 25577^T^ *)
*Cutibacterium granulosum*	NCTC 11865^T^	LT906441 (NCTC 11865^T^)
*Cutibacterium modestum*	JCM 33380^T^	BJEN01 (M12^T^ *)
*Cutibacterium namnetense*	CCUG 66358^T^	LWHO01 (NTS 31307302^T^ *)
*Staphylococcus aureus*	CCUG 1800^T^	CP011526 (DSM 20231^T^ *)
*Staphylococcus capitis* subsp. *capitis*	AYP 1020	CP007601 (AYP 1020)
*Staphylococcus capitis* subsp. *urealyticus*	CCUG 35142^T^	PPQI01 (DSM 6717^T^ *)
*Staphylococcus caprae*	DSM 20608^T^	PPRT01 (NCTC 12196^T^ *)
*Staphylococcus epidermidis*	JCM 2414^T^	CP035288 (ATCC 14990^T^ *)
*Staphylococcus hominis*	JCM 31912^T^	PPQE01 (NCTC 11320^T^ *)
*Staphylococcus lugdunensis*	CCUG 25348^T^	LS483482 (NCTC 12217^T^ *)

## Data Availability

The MALDI–TOF spectra data presented in this study are openly available in the jPOST Repository [https://repository.jpostdb.org/entry/JPST001137, accessed on 7 June 2021], reference number JPST001137 (PXID: PXD025627).

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
