# Peer review of "What Do We See in Spectra?: Assignment of High-Intensity Peaks of Cutibacterium and Staphylococcus Spectra of MALDI-TOF Mass Spectrometry by Interspecies Comparative Proteogenomics"

_microorganisms, 2021, doi:10.3390/microorganisms9061243_

Round 1
Reviewer 1 Report
In manuscript entitled “What do we see in spectra?: Assignment of high-intensity 2
peaks of Cutibacterium and Staphylococcus spectra of MALDI- TOF mass spectrometry by inter-species comparative proteogenomics” Authors have described the proposition of microbial protein (candidates of signal for protein) identification approach for MALDI spectra of Cutibacterium and Staphylococcus. The topic is ambitious and interesting however, in some point is needed additional revision and extended discussion.
This paper should be revised according to the following comments.
- In introduction part Authors should add additional information regarding the nature of molecular profiles of bacteria spectra for linear (mostly positive) mode of bacteria register by using, in case of Authors work, HCCA (CHCA) matrix. The molecular profiles consist of some specific fingerprint, pattern of signals. Right, some 50% of ribosomal, however sometimes results of PSD process, some degradation influence on microbial spectra in linear mode. The artifacts is important to mentioned.
- Utilization of complementary techniques, for improving the identification process is crucial, a specially for close related species. Please, consider it in revision of manuscript. Future Microbiology 15(12):1157-1171
- Please, also discos the influence of extraction procedure for MALDI analysis of microbial profiles in comparison to condition for separation approach such as gel electrophoretic analysis and LC.
- Please, also perform the comparison of gel for NuPAGE with MALDI profiles of bacteria presented in gel view. Perform additional discussion in context of remarks 3.
In my opinion manuscript required the minor revision before final acceptance.
Author Response
Point-to-point response to Reviewer #1
Comment: Authors have described the proposition of microbial protein (candidates of signal for protein) identification approach for MALDI spectra of Cutibacterium and Staphylococcus. The topic is ambitious and interesting however, in some point is needed additional revision and extended discussion… In my opinion manuscript required the minor revision before final acceptance.
Answer: Thank you for your high evaluation that our manuscript is ambitious and interesting, which worth a minor revision to accept. We amended the manuscript based on your and another Reviewer’s comments and made a point-to-point reply as below.
Comment: 1. In introduction part Authors should add additional information regarding the nature of molecular profiles of bacteria spectra for linear (mostly positive) mode of bacteria register by using, in case of Authors work, HCCA (CHCA) matrix. The molecular profiles consist of some specific fingerprint, pattern of signals. Right, some 50% of ribosomal, however sometimes results of PSD process, some degradation influence on microbial spectra in linear mode. The artifacts is important to mentioned.
Answer: Thank you for addressing this important point. We added a sentence in the text Lines 73-79 (line numbers are of Track Changes visible mode), to state PSD in linear mode with HCCA is small enough to ignore as the origin of prominent peaks.
Comment: 2. Utilization of complementary techniques, for improving the identification process is crucial, a specially for close related species. Please, consider it in revision of manuscript. Future Microbiology 15(12):1157-1171
Answer: Thank you for the information. It is remarkable to read that MALDI works better than 16S in some cases. We added the description about this in Lines 55-57.
Comment: 3. Please, also discos the influence of extraction procedure for MALDI analysis of microbial profiles in comparison to condition for separation approach such as gel electrophoretic analysis and LC.
Answer: Thank you for this interesting comment. The extraction procedure is indeed a critical point to evaluate MALDI spectra. In the observation of one of us (I.D.) several years ago, formic acid extraction, which is frequently used in microbial identification, reduces the number of peaks in Cutibacterium and Staphylococcus species. Based on this finding, we performed MALDI-TOF without extraction in this study. However, we can’t present this data as the work was performed in different institution and with non-coauthor colleagues.
In some other genera, it is documented that formic acid extraction increases quality of spectra. However, recent articles focusing this (for example, Cuénod et al., Front Cell Infect Microbiol 2021) look into only the identification accuracy but not the richness of spectra.
Considering these, we feel we can’t discuss the influence of extraction in MALDI-TOF in this manuscript by presenting the actual data or previous studies.
Comment: 4. Please, also perform the comparison of gel for NuPAGE with MALDI profiles of bacteria presented in gel view. Perform additional discussion in context of remarks 3.
Answer: Thank you for the comment. It is not widely understood that MALDI has very high precision compared with gel electrophoresis etc. Therefore, we feel it must be interesting to compare gel photo and MALDI-TOF gel view. We tried to create gel views from our mzXML files. Unfortunately, we couldn’t find any freewares to enable this: Three softwares, which are ClinProTools by Bruker, PVIEW, Pep3D are not distributed anymore, and another software, mzXMLplot, cannot be compiled under Linux environment due to program error.
However, we feel it should be beneficial to add NuPAGE photo in M&M section. We added this as Fig. 1. We hope this is satisfactory.
Reviewer 2 Report
The work entitled:
What do we see in spectra?: Assignment of high-intensity peaks of Cutibacterium and Staphylococcus spectra of MALDI-TOF mass spectrometry by inter-species comparative proteogenomics
Authors: Itaru Dekio *, Yuki Sugiura, Susumu Hamada-Tsutsumi, Yoshiyuki Murakami, Hiroto Tamura, Makoto Suematsu
The scope of work focus on methodology of identification of microbial species and subspecies. The authors utilised proteogenomic approaches to assign the high-intensity peaks of MALDI-TOF spectra of two bacterial genera.
The work give rise to development to achieve more accurate species identification database of MALDI-TOF mass spectrometry based on genome data. However as the author wrote: “As we applied 500 ppm error range, the difference between calculated and observed m/z value is within 3.5 for 7 kD, 5 for 10 kD. This means top-down approach gives several proteins with m/z values close to an observed peak. Therefore, multi-species comparison within a genus is crucial for final verification of the assignment”.
The content of the work is consistent with the proposed title, provided data are clearly described and analyzed. I recommend this work for publication with minor corrections.
Here are my questions concerning scope of the work:
Question no 1:
Page 2
Supplementary materials - Cuti Staph MALDI assignment Suppl ver210426
Table S1. Amino acid sequences of the assigned proteins of Cutibacterium acnes subsp. acnes JCM 6425T 20
The are some discrepancies between observed and calculated values in the table. E.g. in the first case (50S ribosomal protein L7/L12 (rplL)), its molecular weight is 13701.59, while the assigned m/z peak is at 13571.41 for -Met [M+H]+. The mass difference (dM) is 130.18 u, which comes from the contribution brought by methionine as N-terminus amino acid (or valine for 98.12 variation). Please write it clearly otherwise there is a feeling that the values do not match.
Similarly for Lsr2-like protein (dM = 130.19), 10 kDa chaperonin GroS (dM = 98.12), CsbD-like protein (dM = 98.12, based on this value, a more in-depth analysis was then carried out), 7 kD antitoxin (dM = 130.19, based on this value, a more in-depth analysis was then carried out), DUF3117 domain-containing protein (dM = 130.19),
Question no 2:
Page 6, line 234
“3.1. MALDI-TOF peak comparison of Cutibacterium acnes across different oxygen pressures
Three Cutibacterium acnes strains cultured under five different oxygen pressures. The MALDI-TOF peaks of 7 kD antitoxin and CsbD-like protein were observed in all spectra, and the 7 kD antitoxin peak height to CsbD-like protein peak height ratio were lower in lower oxygen pressures (Fig. 2).”
The peaks for 7 kD antitoxin are recorded at the m/z value equal to 7034.5, 7004.6, 6971.5. Why such variation is seen?
The observed and calculated values reported for 7 kD antitoxin-Met [M+H]+ in the subsequent analysis (3.2. MALDI-TOF mass spectrometry profiles of genus Cutibacterium, page 6, line 246 and fig.3b) are much more convergent.
Similarly, the peaks for CsbD-like protein are recorded at the m/z value equal to 7180.0, 7180.0, 7238.0. In the analysis presented in Table S1 CsbD-like protein of MW = 7278.07 was prescribed to m/z = 7179.95 for -Val [M+H]+) and m/z = 3590.48 for -Val [M+2H]2+.
Author Response
Point-to-point response to Reviewer #2
Comment: The work give rise to development to achieve more accurate species identification database of MALDI-TOF mass spectrometry based on genome data. However as the author wrote: “As we applied 500 ppm error range, the difference between calculated and observed m/z value is within 3.5 for 7 kD, 5 for 10 kD. This means top-down approach gives several proteins with m/z values close to an observed peak. Therefore, multi-species comparison within a genus is crucial for final verification of the assignment”.
Comment: The content of the work is consistent with the proposed title, provided data are clearly described and analyzed. I recommend this work for publication with minor corrections.
Answer: Thank you for your high evaluation that our manuscript is consistent with the title and the data are accurately described and analyzed, which worth a minor revision to accept. We amended the manuscript based on your and another Reviewer’s comments and made a point-to-point reply as below.
Question no 1: Page 2. Supplementary materials - Table S1. Amino acid sequences of the assigned proteins of Cutibacterium acnes subsp. acnes JCM 6425T. The are some discrepancies between observed and calculated values in the table. E.g. in the first case (50S ribosomal protein L7/L12 (rplL)), its molecular weight is 13701.59, while the assigned m/z peak is at 13571.41 for -Met [M+H]+. The mass difference (dM) is 130.18 u, which comes from the contribution brought by methionine as N-terminus amino acid (or valine for 98.12 variation). Please write it clearly otherwise there is a feeling that the values do not match. Similarly for Lsr2-like protein (dM = 130.19), 10 kDa chaperonin GroS (dM = 98.12), CsbD-like protein (dM = 98.12, based on this value, a more in-depth analysis was then carried out), 7 kD antitoxin (dM = 130.19, based on this value, a more in-depth analysis was then carried out), DUF3117 domain-containing protein (dM = 130.19),
Answer: Thank you for pointing out this important issue. As you mentioned, it is not possible to follow the calculation flow. Previous reports described posttranslational modifications with N-terminus Met or Val removed occurs in bacterial cells (Tamura 2017 and Belinky 2017, References 3 and 13, respectively). For clarification, we added descriptions in Lines 200-201, 203-204, 206-216 (line numbers are of Track Changes visible mode). I hope these amendments are satisfactory.
Question no 2: Page 6, line 234. Three Cutibacterium acnes strains cultured under five different oxygen pressures.
The MALDI-TOF peaks of 7 kD antitoxin and CsbD-like protein were observed in all spectra, and the 7 kD antitoxin peak height to CsbD-like protein peak height ratio were lower in lower oxygen pressures (Fig. 2).” The peaks for 7 kD antitoxin are recorded at the m/z value equal to 7034.5, 7004.6, 6971.5. Why such variation is seen?
The observed and calculated values reported for 7 kD antitoxin-Met [M+H]+ in the subsequent analysis (3.2. MALDI-TOF mass spectrometry profiles of genus Cutibacterium, page 6, line 246 and fig.3b) are much more convergent.
Similarly, the peaks for CsbD-like protein are recorded at the m/z value equal to 7180.0, 7180.0, 7238.0. In the analysis presented in Table S1 CsbD-like protein of MW = 7278.07 was prescribed to m/z = 7179.95 for -Val [M+H]+) and m/z = 3590.48 for -Val [M+2H]2+.
Answer: Thank you for the question. These differences between C. acnes subtypes derive from protein sequence differences engraved in the gene. We gained the gene sequences from their genome and calculated the m/z values based on them. For clarification, we added the Figure 3B to visualize the protein sequence differences across C. acnes subtypes and Cutibacterium species that lead to m/z value differences.
Reviewer 3 Report
The authors presented a manuscript describing a biotyping analysis of two species of bacteria, including several subspecies. Combination of a top-down, bottom-up, and mass calculation based on genomic data is a good approach to annotate MS peaks. My enthusiasm for the manuscript was lessen without a clear rational for the annotation of the most intense peaks. Selection of the most abundant peaks were matched to the mass calculation from publicly available genomic data, however the inclusion of bottom-up data only focused on two proteins in the text. Little details were provided regarding the confidence of annotation from the bottom-up approach. Based on the number of proteins within a m/z bin and the relatively large mass error in mass match (500-600ppm), additional information should be provided in the supplemental to support the annotations (e.g., how many peptide hits were observed for each protein annotation).
Lines 41-44, the bacterial isolate is not crystallized but the matrix is crystallized. The laser shot produces ions, where the ionized matrix transfer ionization to ablated analytes. Reducing the MS to a vacuum tube is an oversimplification.
Lines 46-48, consider rephasing, sentence is unclear. The authors seem to be describing that the profile is very useful while individual peaks are unknown. Authors should clarify how the identification of individual peaks would improve the process. Include rationale why most dominate peaks should be the focus.
Line 60-63, considered rephasing, sentence is unclear. Authors should try to explain the magnitude of unannotated peptides in relationship to all detected.
Include the size of the laser shot and number of laser shots in the analysis. If more than one shot, was there a pattern to the laser shots or random?
Since three spectra were collected from presumably three different colonies, statistics to show reproducibility would be useful to understand homogeneity of the culture, perhaps authors may have sampled colonies at different phases.
Lines 168-169, samples are not “applied to” rather analyzed with.
m/z should be italicized.
For peaks 15043 and 7522, in supplemental the authors have a case that the peaks may be related to hemoglobin alpha chain from sheep’s blood. Unclear why present the peaks as unknowns instead of impurity from culturing further support by the elimination of sheep blood from culture conditions. A supplemental figure capturing the spectral difference would be useful, with the assumption that there are other spectral differences.
Supplemental tables lack explanation for underlined or double underlined entries.
Figure 3B and 4C label axis.
Round 2
Reviewer 3 Report
Line 47, change “crystallization by applying “ to “application of”. Also change “, then…” to “, then a LASER desorbs the crystallized matrix and sample to form a cloud..”
Lines 414-417, Unclear what was meant in “… a verification based on species difference was impossible.” Comparison to a standard should be sufficient.
Author Response
Point-to-point response (Revision 2) to Reviewer #3
Comment: Line 47, change “crystallization by applying “ to “application of”. Also change “, then…” to “, then a LASER desorbs the crystallized matrix and sample to form a cloud..”
Answer: Thank you very much for indicating these points to improve our manuscript. We amended the text in Lines 48-50 (line numbers are of Track Changes visible mode).
Comment: Lines 414-417, Unclear what was meant in “… a verification based on species difference was impossible.” Comparison to a standard should be sufficient.
Answer: Thank you very much for this comment. We agree with your thought and amended the text in Line 424. We hope you feel this change is satisfactory.